# Post-Translation Modifications and Mutations of Human Linker Histone Subtypes: Their Manifestation in Disease

**DOI:** 10.3390/ijms24021463

**Published:** 2023-01-11

**Authors:** Ashok Kumar, Preeti Maurya, Jeffrey J. Hayes

**Affiliations:** 1Department of Biochemistry and Biophysics, University of Rochester, Rochester, NY 14642, USA; 2Aab Cardiovascular Research Institute, University of Rochester, Rochester, NY 14642, USA

**Keywords:** linker histone, post-translational modification, H1 subtypes, H1 PTMs and disease

## Abstract

Linker histones (LH) are a critical component of chromatin in addition to the canonical histones (H2A, H2B, H3, and H4). In humans, 11 subtypes (7 somatic and 4 germinal) of linker histones have been identified, and their diverse cellular functions in chromatin structure, DNA replication, DNA repair, transcription, and apoptosis have been explored, especially for the somatic subtypes. Delineating the unique role of human linker histone (hLH) and their subtypes is highly tedious given their high homology and overlapping expression patterns. However, recent advancements in mass spectrometry combined with HPLC have helped in identifying the post-translational modifications (PTMs) found on the different LH subtypes. However, while a number of PTMs have been identified and their potential nuclear and non-nuclear functions explored in cellular processes, there are very few studies delineating the direct relevance of these PTMs in diseases. In addition, recent whole-genome sequencing of clinical samples from cancer patients and individuals afflicted with Rahman syndrome have identified high-frequency mutations and therefore broadened the perspective of the linker histone mutations in diseases. In this review, we compile the identified PTMs of hLH subtypes, current knowledge of the relevance of hLH PTMs in human diseases, and the correlation of PTMs coinciding with mutations mapped in diseases.

## 1. Introduction

The eukaryotic nuclear DNA is packaged inside the nucleus in a beads-on-a-string structure by wrapping ~147 bp around the core histone (H2A, H2B, H3 and H4) octamer to form nucleosome cores, connecting the cores by stretches of linker DNA. Linker histones (LH) bind at the linker entry and exit site on the nucleosome to stabilize and condense the chromatin to form higher-order chromatin fiber [1,2,3]. The condensation and decondensation of the chromatin by LH regulates the dynamic function of cells such as cell cycle, replication, DNA repair, RNA turnover, transcription and development [4,5,6,7,8]. The LH functional diversity thought to be augmented further in higher eukaryotes because of simultaneous expression of multiple variants (11 subtypes): seven somatic (H1.0, H1.1–H1.5 and H1.10) and four germ cells (H1T, H1T2, HILS1 and H1OO). The sequence variations in the subtypes indicates conserved distinct structural and functional properties of these subtypes in chromatin arrangement and cellular processes [9]. Since the discovery of H1s, numerous studies have been performed to understand the redundant and non-redundant biological roles of H1s in cellular processes. For example, deletion of one or two subtypes is compensated by the overexpression of others, while the deletion of three subtypes is found to be embryonically lethal in mice [10,11,12,13]. Furthermore, a diverse array of post-translational modifications (PTMs) in H1s adds to the potential complexity of LH diverse functions, but the functions of most LH PTMs are poorly understood [14].

Epigenetics play a crucial role in regulating the physiological processes of the cell and are influenced by environmental stimuli. Epigenetic modifications, in general, are reversible phenomena and have been associated with many pathologies, including several cancers, and therefore, the processes regulating these modifications have become drug targets [15]. As per the histone code hypothesis, the different modifications of histone acting alone or simultaneously have distinct downstream functions [16,17,18]. Advancements in mass spectroscopy have revolutionized the precise identification of PTMs in proteins and have led to the identification of a number of PTMs in LH subtypes, compared to the cumbersome methodology of radioactivity and antibody-based detection. The PTMs of canonical histones are identified unambiguously and have been extensively studied, while the PTMs of LHs are beginning to be identified in initial studies, in nine out of the eleven mammalian subtypes [19]. LHs have a net basic charge, and their CTD is rich in basic charged residues, especially lysine, with few arginine residues, which promotes chromatin condensation and functions in diverse cellular processes such as DNA damage, transcription and cellular differentiation [20,21,22,23,24,25]. In addition to PTMs, cancer genome sequencing studies have highlighted mutations in a number of the proteins involved in regulating epigenetic marks, as well as in histone proteins themselves, indicating the importance of these marks in disease [26,27,28]. In this review, we compare the identified PTMs of human H1 subtypes, discuss what is known regarding their functional relevance in human disease, and also describe mutations of human LH (hLH) subtypes found in cancers.

## 2. Human Linker Histone and Its Subtypes

LHs in higher eukaryotes have a tripartite structure, with a trypsin-resistant central globular domain (GD) (~80 residues) flanked by unstructured and protease-sensitive N-terminal domain (NTD) (~25–35 amino acids), and C-terminal domain (CTD) (~100 residues) (Figure 1A). The CTD possesses more than 40 lysine residues, thus making it a highly positively charged segment of LH. Indeed, while the structured GD is responsible for structure-specific recognition and binding to the nucleosome surface, the CTD stabilizes binding via interaction with the linker DNA segments (Figure 1B), as well as stabilizing higher order chromatin structure by neutralizing the densely packed negative charge of DNA [1,2,29,30,31,32]. The wing helix domain (WHD) of the LH interacts with the nucleosome over the dyad axis and with the first ~10 bp of each linker DNA in a symmetric or asymmetric fashion suggested to be based on chromatin dynamic events [33,34,35]. Linker histones are evolutionarily more diverse than the canonical histones, but the GD is conserved through evolution in plants, fungi and animals and exhibits the greatest sequence conservation, while the NTD and CTD exhibit greater sequence variation [36]. Initially, few LH subtypes had been reported [37]; however, with extensive research, multiple subtypes of LHs have been reported across different species: e.g., *Gallus gallus* with 7, *Drosophila melanogaster* has one somatic and one embryonic subtype, *Saccharomyces cerevisiae* has one, and mammals have 11 subtypes. Interestingly, LH subtypes have more similarity between species when compared within the species (Figure 1C) [36,38]. LHs are also categorized based on their expression patterns in tissues. For example, humans have two classes of somatic subtypes: five that are ubiquitously expressed in a replication dependent fashion (H1.1-H1.5), and two that are expressed in a replication independent, and found mainly in terminally differentiated cells (H1.0 and H1x). Humans also have germ cell-specific LHs, with one oocyte specific (H1oo), and three testis specific (H1T/H1.6, H1T2, and HILS1). These subtypes are distributed over a wide variety of genomic locations, with H1.1, H1.2, H1.3, H1.4, H1.5, H1.0, H1X/H1.10, H1T/H1.6, H1T2/H1.7, H1LS1/H1.9, H1oo/H1.8 found on chromosomes 6p21.3, 6p21.3, 6p21.3, 6p21.3, 6p22.1, 22q13.1, 3q21.3, 6p21.3, 12q13.1, 17q21.33, 3q22.1, respectively [39,40,41]. Although subtypes within the species show higher similarity in their GD sequence, they possess heterogeneity among amino and carboxyl terminal domain sequences. In addition, there have been a number of PTMs reported in these domains. Overall, the heterogeneity in their NDT, CTD and PTMs can be considered for diversity in LH functions.

## 3. Post-Translational Modifications of Human Linker Histones

LHs are an essential component of chromatin and play diverse roles in cellular processes. The functional diversity of LHs are because of the heterogeneity in their unstructured N and C-terminal domains and the PTMs in these domains. The PTMs of canonical histones were first identified in 1960s and extensively studied, while the first PTM of LH was identified almost a decade later, within *Physarum polycephalum* H1 using radioactive ^32^P labelling followed by electrophoresis and, successively, from different species *Drosophila melanogaster*, chicken erythrocytes Chinese hamster ovary cells, and mammalian cell lines [19,42,43,44,45,46]. However, because of ^32^P labelling, difficulties in separation of the linker histone subtypes, the lack of subtype-specific antibodies, and the lack of a highly sensitive method to detect the PTMs in low amounts of proteins, such LH PTMs studies were limited. The introduction of mass spectroscopy, trypsin digestion and separation of the peptides by HPLC followed by amino acid analysis using the Edman degradation helped in the identification of the phosphorylated LH. However, the exact site of phosphorylation in LH was elusive for many subtypes until advancements in PTMs identification methodologies. The technical limitation of retention of highly positive charge (basic) peptides on C18 resin (capillary column) were overcome by using a propionylating reagent which removes the charge from unmodified and monomethylated lysine residues and adds a hydrophobic propionyl group which increases the retention time of the peptide on the C18 column. In addition, the combination of high-performance liquid chromatography (HLPC), reversed-phase (RP)-HPLC, hydrophilic interaction liquid chromatography (HILIC), high-performance capillary electrophoresis (HPCE), enzymatic cleavage, amino acid sequence analysis, and linear quadrupole ion trap-Fourier-Transform Ion Cyclotron Resonance (LTQ-FT-ICR) mass spectrometry provided better coverage, high resolution and mass accuracy, leading to the identification of LH subtypes [47]. Furthermore, LH subtype separation and additional PTMs (ubiquitination and formylation) were detected with high confidence from HeLa and MCF7 cell lines using liquid chromatography (LC) connected to an LTQ-Orbitrap mass spectrometer having a nanoelectrospray ion source [48]. Starkova et al. employed Matrix-activated laser desorption/ionization Fourier-transform ion cyclotron resonance mass-spectroscopy (MALDIFT-ICR-MS) coupled with acetic acid–urea polyacrylamide gel electrophoresis (AU-PAGE), second dimension SDS-PAGE and trypsin digestion, which led to the separation of LH subtypes (H1.0, H1.X, H1.1, H1.2, H1.3, H1.4 and H1.5) from human K562 cell line from tissue samples of mouse and calf thymi to separate the LH subtypes (H1.1, H1.2, H1.3, H1.4) and identification of novel PTMs at meK75-hH1.3, acK26-hH1.4, acK26-hH1.3 and acK17-hH1.1 [49].

The NTD and CTD of many LH subtypes contains multiple cyclin-dependent kinase (CDK) motifs ((S/T)PXZ, where X is any amino acid and Z is a basic amino acid), and in vivo data suggests that these CDK motifs undergo site-specific cell-cycle-dependent phosphorylation [20,50]. Mass spectroscopy and biochemical studies established that basic amino acids of LH are primarily responsible for its interaction with chromatin and their stabilization. The LH PTMs such as phosphorylation, acetylation, formylation, propionylation, and crotonylation result in reduction of the positive charge of LH, which impacts the chromatin stability and furthermore the compact state of the chromatin, while methylation tends to be associated more repressed chromatin state [20,49,50,51]. An overview of LH PTMs in cellular function is shown in Figure 2. In this section, we compile the known PTMs of LHs that have been identified in humans, excluding other mammalian species, as differences in species or even model systems may impact the LH PTM differently. The most commonly identified human LH PTMs are phosphorylation, acetylation, and methylation, with less extents of formylation, citrullination, sumoylation, and lysine β-hydroxybutyrylation, as identified by mass spectrometry of proteins from different cell lines and clinical samples. The PTMs of hLH subtypes are tabulated in Table 1 as per the translating protein sequences. It can be noticed from the table that although PTMs are distributed across the hLH subtypes sequences, some lysine residues within the GD can be modified with several different PTMs at the same position [4,52,53,54,55,56,57,58].

## 4. Modulators of Linker Histone Post-Translational Modifications

The emerging role of linker histone PTMs in cellular processes generates interest in how these PTMs are modulated. Phosphorylation is one of the extensively studied PTMs of LH and has been found to be cell cycle dependent with the lowest level in G1, reaching to maximal level in metaphase [47,59]. It has been found that human H1.4S27, H1.4S35 and H1.5T10 are phosphorylated by Aurora B kinase, protein kinase A, and glycogen synthase kinase-3, respectively, and they are predominant in the mitotic stage, while human H1.5S17, in phosphorylated form, is observed in early G1 phase, and H1.5S172 and H1.5S188 phosphorylation occurs in S phase. Although protein phosphatase 1 (PP1) has been suggested as a phosphatase for reversing the LH phosphorylation, the protein kinase inhibitor staurosporine has been reported to specifically dephosphorylate H1.5T10 compared to H1.5S17, H1.5S172 and H1.5S188, and prolonged dexamethasone exposure (48 h) in Mouse 1471.1 cells reduces the phosphorylation level of H1.3 and H1.4 [14,21,22,59,60,61]. In addition, positive transcription elongation factor b (P-TEFb), the complex of cyclin T1 and CDK9 reported to phosphorylate the LH and inhibition of P-TEFb by RNAi, flavopiridol, or dominant negative CDK9 expression results in a reduction in phosphorylation. Furthermore, H1.4K26 is methylated by either Ezh2 or G9a, and can be demethylated by members of the Jumonji domain 2 (JMJD2) subfamily of demethylases, similar to an analogous sequence in the N-terminal tail domain of H3. However, methylation of H1.2 at K187 is mediated by G9a in association with its binding partner Glp1 and H1.2K187 methylation is non reversible by the JMJD2. Acetylation of H1.4K34 is associated with GCN5 while H1.4K34, and H1.4K26 gets deacetylated by SIRT1. The writer of H1.4K26 acetylation still need to be explored [14]. In DNA damage condition, H1.2 S188 PARylation is mediated by PARP1, which poly-ADP-ribosylate it and in turn impacts the association with chromatin [62]. Peptidylarginine deiminase 4 (PADI4) has been suggested to mediate the citrullination at R54 of LH subtypes (H1.2, H1.3 and H1.4) [25]. The understanding of the modulators of hLH PTMs can help in finding novel small molecule inhibitors or activators such as dexamethasone, flavopiridol, and staurosporine.

### 4.1. Linker Histones PTMs in Disease

Epigenetic marks either on DNA or protein have been of great interest to the scientific community. In recent years, the histone PTMs have drawn attention for extensive investigation in human disease. A number of studies have established a connection between PTMS within core histone in cancer progression and response to therapy as well as potential as a biomarker of cancer. It has been reported in many cancerous cell lines and tissue types that hyperacetylation of histone H4 acetylation at lysine 16 and trimethylation at lysine 20 occurs early during tumorigenesis and, hence, indicates a general epigenetic mark of cancer [63,64,65]. Moreover, in line with this, LH has also drawn attention, and differential expression and level of LH subtypes has been found altered in cancerous samples in addition to the alteration of epigenetic marks [28]. Overall levels of H1 subtypes have been linked with tumor grade and aggressiveness of cancers. For example, high expression of H1.5 has been reported in high-grade tumors (pulmonary neuroendocrine tumors and prostate cancer), and reduced expression with low-grade tumors [66,67]. H1.0 is highly expressed in terminally differentiated cells, and its expression is generally decreased in various cancers, and marks cancer stem cells [68]. In triple-negative breast tumors and ovarian cancers, it has been found that levels of LH subtypes H1.0, H1.1, H1.4 and H1x are reduced, and they are also decreased in malignant adenocarcinomas compared with benign adenomas [54,69]. However, higher levels of H1.2 in tumors appear to be correlated with poor outcomes in pancreatic cancers [70,71]. A detailed knowledge of PTMs associated with LH subtypes in cancers is of great interest in understanding how epigenetic mechanisms contribute to cancer phenotypes. We refer readers to several reviews that explore the diverse functions of LH and their PTMs in various cellular processes [14,52,55,56,72] and the differential expression profile of hLH subtypes in cancers [28,73]. In this section, we focus on compiling information from the literature and databases related to the role of hLH PTMs in disease.

The phosphorylation of T146 of H1.2, H1.4 and H1.5 in human noninvasive low-grade (RT4) and invasive high-grade (T24, J82, and UMUC3) bladder cancer cell lines has been identified in a cell cycle-dependent manner and suggests this modification correlates with bladder cancer development and progression [74]. However, a study of liver cancer cell lines (HuH6, SNU449, and THLE-2 cells) and normal cell lines has found that metastasis-associated 1 protein (MTA1) reduces the phosphorylation of H1.2 at T146 by proteasomal degradation of DNA-PK, which in turn promotes hepatocellular carcinoma (HCC). In addition, immunohistochemical staining of 242 breast cancer tumor samples and 97 non-tumor samples with H1 T146p-antisera indicates that this PTM has the potential to serve as a biomarker to detect HCC, breast cancer, and bladder cancer. Possible ‘writers’ for the phosphorylation of T146 are Aurora B Kinase, Protein kinase A (PKA), and Glycogen synthase kinase-3 beta (GSK3) [56,75,76,77].

Combining 2D-TAU/SDS gel electrophoresis with LC-MS/MS led to the identification of novel tyrosine phosphorylation sites in three hLH subtypes in MCF7 and HCC1937 breast cancer cell lines. Focal Adhesion Kinase (FAK) has been identified as the putative writer for this phosphorylation which occurs at Y74 of H1.5, Y70 of H1.2 and Y71 of H1.3. The level of tyrosine phosphorylation increases in a cell cycle-dependent manner as the FAK translocates from the cytosol to the nucleus, exhibiting low levels in interphase which increase in metaphase, telophase and anaphase. The phosphorylation level of (Y70p-H1.2, Y71p-H1.3 and Y74p-H1.5) in breast cancer cells is high compared to normal cells, and suggests is a role for this LH modification in breast cancer. In addition, it has been determined that the addition of epidermal growth factor (EGF) can increase LH tyrosine phosphorylation, perhaps acting as a kind of cofactor. Of note, the treatment of MCF7 and HCC1937 cells with an FAK inhibitor (PND1186), PI3K inhibitor (LY294002) or a peroxisome proliferator-activated receptor γ (PPARγ) agonist (troglitazone) significantly decreased the tyrosine phosphorylation levels. P13K downstream signaling targets such as PPARγ and AKT have been found to regulate the level of FAK, which in turn impacts the phosphorylation of H1 subtypes. Overall, the novel tyrosine phosphorylation of these subtypes by FAK indicates a nexus of signaling, which may have broader impact in other diseases in addition to breast cancer [58].

The hLH H1.4 is acetylated by General Control Non-depressible 5 (GCN5) at K34. The acetylated K34 is augmented at the promoters of transcriptionally active genes, regulatory regions and found to interact with TAF1 subunit to recruit the TFIID to the promoter in order to facilitate active transcription. In addition to multiple cellular functions such as transcription, chromatin binding, and spermatogenesis, H1.4 K34ac has been linked with seminomas. The screening of H1.4 K34ac levels in testicular germ cell tumors (malignancies more frequent in Caucasian males) in the biopsies of carcinomas in situ and seminomas showed increased levels of H1.4 K34ac, implicating a possible role of this modification in these cancers [23].

Squamous cell carcinoma of the head and neck (SCCHN) includes cancers of the nasal cavity, mouth, salivary glands, sinuses, lips, throat, and larynx. It has been reported that in Squamous cell carcinoma cells SCC-35, the Wolf–Hirschhorn syndrome candidate 1 (WHSC1) protein, a lysine methyltransferase, interacts with LHs and monomethylates lysine 85 in the globular domain of LHs H1.1, H1.2, H1.4 and H1.5, and is found to be associated with all replication-dependent somatic hLH subtypes. The K85 residue is conserved among the somatic LH subtypes except H1x (note that the corresponding residue in H1.1 and H1.5 is K84, while in H1.3, it is K86). The H1.4 K85A mutational study revealed that monomethylation is involved in regulating gene expression and cell proliferation, mediated through OCT4 stemness [78]. In another study, H1.4K 85 acetylation by the acetyltransferase P300/CBP-associated factor (PCAF) regulates chromatin structure by interacting with the heterochromatin protein HP1, and in the case of DNA damage H1.4Km85ac leads to chromatin relaxation and genomic stability [79]. Overall, K85 of LH is a critical residue involved in chromatin and gene regulation, likely due to effects on GD binding, and mutation or PTMs at this site impact the normal cellular function.

Surprisingly, non-nuclear LH has been correlated in neurodegenerative disease such as Alzheimer’s disease (AD) and Parkinson’s disease (PD) [80,81]. It has been reported that protein kinase Cdk2 and Cdk5 phosphorylate the linker histone at the CDK motif site and destabilize its interactions with chromatin, which in turn leads to cytoplasmic localization of LH [81,82,83]. It has been postulated that this cytosolic LH interacts with intracellular amyloid proteins, i.e., amyloid-β and α-synuclein, in AD and PD conditions. Furthermore, in vitro assays such as fluorescence spectroscopy, pool down, electron microscopy, and surface plasmon resonance (SPR) spectroscopy assay confirm the LH association with amyloid-β and α-synuclein proteins [81,84,85] and highlight the role of LH in neurodegenerative disease. However, we are still missing the unblemished evidence of critical residues of LH subjected to post-transnational modifications in the neurodegenerative disease.

### 4.2. Mutations in Linker Histones and Their Implications

The whole-genome sequencing of clinical samples revealed high-frequency mutations which have been reported for the linker histone subtypes in the COSMIC database. Analysis of this data suggests that mutations are associated with multiple cancers (e.g., pancreatic, cervical, head and neck and colorectal cancers). In this section, we are describe the percent distribution of numerous mutations analyzed from the COSMIC database in pie chart form (Figure 3). The chart shows missense mutations have the highest distribution among all the subtypes.

In addition, a novel frameshift mutation in human H1.4 subtypes has been reported. The frameshift mutation in H1.4 results in a rare congenital anomaly syndrome known as Rahman syndrome, which is characterized by overgrowth, distinctive facial features, intellectual disability, premature aging and behavioral problems. The frameshift mutations which involve deletion or insertion end up in a truncated protein with a drastic reduction in the overall net positive charge of human H1.4 compared to the WT protein. The decrease in positive charge in turn impacts the DNA binding, chromatin stability, chromatin remodeling, and gene expression ability of the protein. There are reports of more than 20 types of frameshift (for example, at K173, K139, T142, K148, S150 and K157, etc.) mutations in human H1.4 subtypes from the patient’s samples that have been identified, which result in early truncation of CTD and the formation of a pathogenic variant of this protein with abnormal function that cause of Rahman syndrome [86,87,88,89]. The expression of an H1.4 CTD-truncated mutant in rat revealed the down-regulation of nearly 400 genes, and it has been found that many of these genes are involved in neuronal activities [86].

## 5. Future Prospective

While research of linker histones has been progressing since their discovery, has did not matched the pace of studies of the core histones. However, there have been a number of landmark contributions elucidating the role of LHs in chromatin stabilization, 3-D structure, transcription, DNA replication, and repair. Furthermore, suggestions of redundant functions of linker histone subtypes have become more delineated, and the functional contribution of the subtypes, especially somatic linker histone subtypes, has been explored. The advancement in mass spectrometry led to the identification of PTMs in its subtypes and, further, the functional significance of the PTMs in chromatin and cellular processes such as apoptosis. Furthermore, genome sequencing in the tissue of cancerous patients’ samples has led to the outcome of numerous high-frequency mutations in hLH subtypes, indicating the potential of linker histone subtypes in multiple cancers. In line with this, the whole-exome sequencing (WES) of a 10-year-old boy with autism and intellectual disability has been identified with a deleterious mutation in linker histone H1.4 (HIST1H1E; c.435dupC; p.Thr146Hisfs*50). The frameshift mutation, or substitution of T146 to T146H results in the early truncation of this protein and therefore proposed to impact the overall function of H1.4 [86]. We note that T146 is also the site of phosphorylation of the hLH H1.4 subtype and is conserved among the hLH somatic subtypes (H1.2–H1.5), as mentioned earlier, indicating further the critical nature of thisese amino acids and the possible role of PTMs in diseases [74]. The role of PTMs and the genetic mutations at the same site have been noticed in actin as well, and we recently identified a novel conserved nuclear actin-R256me1 PTM using a yeast model system which was not identified earlier because of the technical limitations in getting a pool of nuclear actin for PTMs studies. In addition, we established the role of this epigenetic mark (actin-R256me1) in the nuclear process and possible implications of actin-R258 (analogous in human) mutation in thoracic aortic aneurysms and dissections (TAAD) and in Moyamoya-like cerebrovascular disease [90,91]. Similarly, a heterozygous mutation (AAA to AGA) has been reported in Raji B lymphoblastoid cells, lysine at position 173 (K173R) in hLH H1.4 subtype. The frequency of this mutation is 6.3% among the Swedish population [47]. Although the K173 has not been identified with PTMs, to the best of our knowledge, but this region falls in the SPKK motif, and being in the C-terminal segment, the possibility of its PTMs cannot be ruled out. It may be possible that careful analysis and technical advancement in the future will reveal PTMs at this position, similarly to how we have identified novel actin-R256me1. Moreover, there can be many more critical PTMs and PTMs coinciding with genetic mutations that might appear in future studies with mechanistic detail by the advancement in handling the clinical samples using mass spectrometry and genome sequencing and, furthermore, by developing the novel PTMs of hLH subtypes and site-specific antibodies.

## Figures and Tables

**Figure 1 ijms-24-01463-f001:**
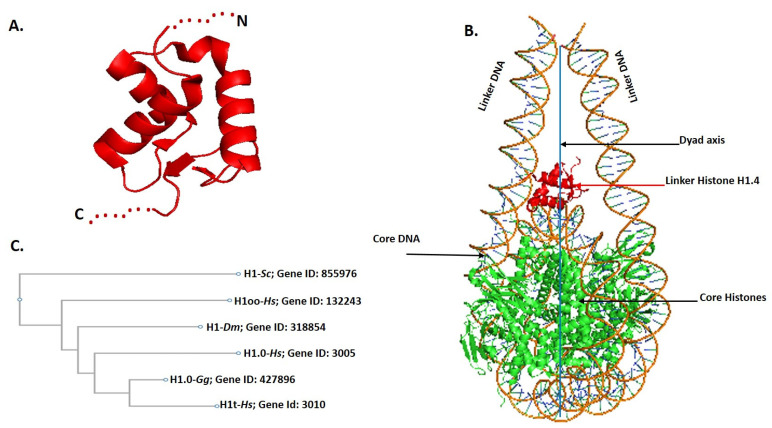
(**A**) General structure of linker histone with central globular domain (red), unstructured N-terminal domain (dotted red), and long C-terminal domain (dotted red). (**B**) CryoEM structure of the nucleosome with linker histone H1.4 prepared by PyMol using PDB ID: 7k5y. The central portion is octamer of histone (green), Linker histone H1.4 (red). (**C**) Phylogeny tree of linker histone from *Saccharomyces cerevisiae* (*Sc*), *Homo sapiens* (*Hs*), *Drosophila melanogaster* (*Dm*), *Gallus* (*Gg*).

**Figure 2 ijms-24-01463-f002:**
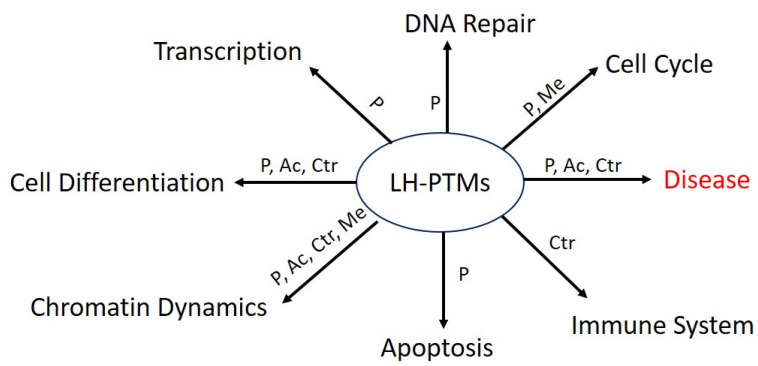
**A general overview of involvement of linker histone PTMs in cellular processes;** Linker histone PTMs (phosphrorylation (P), acetylation (Ac), methylation (Me), and citrulation (Ctr) have critical involvement in the normal physiological and disease states.

**Figure 3 ijms-24-01463-f003:**
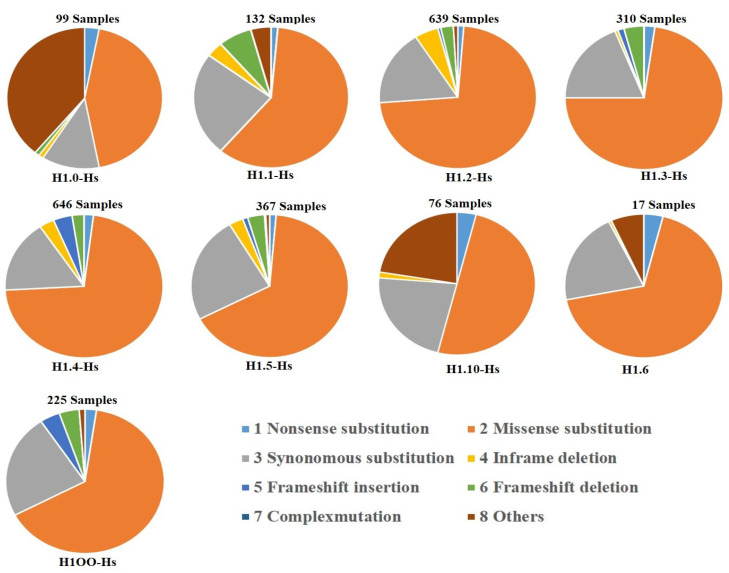
**Mutations in Human Linker histone:** The pie chart derived from information available at COSMIC database (https://cancer.sanger.ac.uk/cosmic) on 30 October 2022 indicates various mutations observed in different cancerous tissue of the patient samples. The frequency of distribution of mutations are indicated in percentage.

**Table 1 ijms-24-01463-t001:** Post-translation modifications of human linker histone subtypes.

Human Linker Histone Subtypes	Phospho-rylation	Acetylation	Methyltion	Formy-lation	Ubiquitn-ylation	Hydrox-ylation	2-hydroxyisob-utyrylation	Citrulation	Crotony-lation	ADP-ribosylat-ion (Parylation)	Refrences
H1.1	S2, S115, S116, T119, T130, T133, S136	K17, K22, K55, K67, K78, K88, K93, K100, K137, K185, K186, K187, K191, K193	K67, K93, K100, K109, K113, K120, K131, K140, K188, K191, K193, K196, K208, K213	K67, K84, K93, K100	K78, K100	Y74, K100					[4,46,48,49]
H1.2	S2, T4, T31, S36, T45, Y71, T146, T154, T165, S173	K17, K23, K34, K46, K52, K63, K64, K85, K90, K97, K148, K169, K172, K175, L176, K178, K183, K192	K27, R33, K34, K46, K52, K63, K64, K75, K85, K90, K97, K106, K119, K129, K148, K168, K172, K181, K183, K187, K196, K201, K211	K17, K34, K46, K63, K64, K75, K81, K85, K90, K97, K160	K34, K46, K63, K64, K75, K85, K90, K97, K106	Y71	K45, K51, K62, K63, K84, K89, K96	R54	K34, K64, K85, K90, K97, K159	S188	[4,14,46,47,48,49,53,56,57,58]
H1.3	S2, T4, T18, T30, S37, Y72, T147, T155, T180, S189	K17, K25, K26, K35, K47, K53, K64, K65, K86, K91, K98, K138, K140, K141, K149, K169, K179, K184, K185, K188	K25, K26, K33, K35, K47, K53, K64, K65, K76, K91, K98, K107, K118, K137, K149, K150, K169, K170, K173, K194, K204	K35, K47, K64, K65, K75, K82, K86, K91, K98, K141, K160	K35, K47, K65, K76, K86, K91, K98, K107	Y72		R55		S189	[4,14,46,47,48,49,53,56,58]
H1.4	S2, T4, T18, S27, S36, S41, T142, T146, T154, S172, S187	K17, K26, K32, K34, K46, K52, K63, K64, K85, K90, K97, K169, K190, K192	K17, K21, K22, K23, K25, K26, K32, K33, K34, K46, K52, K63, K64, K75, K85, K90, K97, K106, K119, K121, K127, K136, K148, K159, K168, K169, K177, K192, K195, K197, K207, K212, K217	K17, K34, K46, K63, K64, K75, K81, K85, K90, K97, K110, K140, K160	K17, K21, K34, K46, K63, K64, K75, K85, K90, K97, K106	Y71		R54		E3, E16, E115, K219	[4,14,46,47,48,49,53,56,57]
H1.5	S2, T4, T11, S18, S36, T39, S44, Y74, S107, T138, T155, S173, T187, S189	K17, K49, K67, K78, K88, K93, K109, K168, K169, K209	K27, K49, K55, K67, K78, K84, K93, K100, K109, K130, K149, K168, K169, K172, K182, K194, K199, K214, K224	K37, K67, K84, K88, K93	K35, K37, K49, K100			R57			[4,14,46,47,48,49,53,56,58]
H1.0	T2		K12, K82, K102, K108, K155		K40			R43, R94			[4,14,46,47,48,49,53,56,58]
H1X	S2, S31				K106						[46,48]

## Data Availability

Not applicable.

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
