# Peer review of "Post-Translation Modifications and Mutations of Human Linker Histone Subtypes: Their Manifestation in Disease"

_ijms, 2023, doi:10.3390/ijms24021463_

Round 1

Reviewer 1 Report

In this review by Kumar and Maurya the authors have discussed the post-translation modifications and mutations of human linker histone subtypes and have correlated with various diseases. Overall, it is an extensive review of literature, and the sections are organized in a sequential manner. Here are my suggestions to improve this review.

1. Abstract, please remove “up to some extent”.

2. Throughout the manuscript change all “a.acid to amino acids”.

3. In page 2, please give a phylogenetic tree with gene id and schematic domains of linker histones for chicken, drosophila, saccharomyces, and human/mammal to strengthen the sentence “Chickens possess seven, Drosophila melanogaster one 81 somatic, Saccharomyces cerevisiae one, and mammals 11 subtypes LH subtypes have 82 more similarity between the species when compared within the species”.

4. Fig 1 A is not very scientific. Either do a good schematic or use the PyMol to show the Nt-GD-Ct.

5. Table 1: Please give references.

6. Page 5 typo “Epigenetic marks” e is missing.

7. Page 6 line 162 put space “profile of”

8. Page 7, line 236- zk139?

9. Page 8, line 265, correct T146H or write clearly.

10. A paragraph must be devoted to technological evolution to LC-MS in identification and detection of LH modification. With a brief historical measurement, the shortcomings and LC-MS efficiency. This will make the review more complete.

Author Response

Reviewer#1

In this review by Kumar and Maurya the authors have discussed the post-translation modifications and mutations of human linker histone subtypes and have correlated with various diseases. Overall, it is an extensive review of literature, and the sections are organized in a sequential manner. Here are my suggestions to improve this review.

We are highly thankful to the reviewer for their time and appreciating the arrangements of the sections of the manuscript and providing their constructive remarks.

  1. Abstract, please remove “up to some extent”.

Thank you for noticing it, we rephrased the whole abstract.

  1. Throughout the manuscript change all “a.acid to amino acids”.

As per suggestion a.acid to amino acids has been replaced in the revised version of the manuscript.

  1. In page 2, please give a phylogenetic tree with gene id and schematic domains of linker histones for chicken, drosophila, saccharomyces, and human/mammal to strengthen the sentence “Chickens possess seven, Drosophila melanogaster one 81somatic, Saccharomyces cerevisiae one, and mammals 11subtypes LH subtypes have 82 more similarity between the species when compared within the species”.

We are thankful for the reviewer suggestion and added the phylogenetic tree as Figure 1a. For the sake of simplicity we chose one linker histone from chicken (Gallus gallus, and one somatic, one testis specific and oocyte specific from Homo sapiens) in the figure. It can be seen from the figure that H1 from Saccharomyces is closer to Homo sapiens H1oo and Gallus gallus is closer to Human H1t-Hs (testis specific subtypes).

  1. Fig 1 A is not very scientific. Either do a good schematic or use the PyMol to show the Nt-GD-Ct.

In our revised manuscript, we did rearrangement of our Figure1 and figure 1a is replaced by a phylogenetic tree figure and next added the cartoon representation/diagrammatic for LH (Figure 1b) made by using PyMol as per reviewer suggestion.

  1. Table 1: Please give references.

We agree with the reviewer and thank you for pointing us to this. We replaced the older Table 1 with new Table 1 with a column of the cited references.

  1. Page 5 typo “Epigenetic marks” e is missing.

We corrected the typo error in the revised manuscript.

  1. Page 6 line 162 put space “profile of”

We added the space and thankful to the reviewer for noticing it.

  1. Page 7, line 236- zk139?

We fixed this typo error in the revised manuscript.

  1. Page 8, line 265, correct T146H or write clearly.

This error is corrected in the revised manuscript.

  1. A paragraph must be devoted to technological evolution to LC-MS in identification and detection of LH modification. With a brief historical measurement, the shortcomings and LC-MS efficiency. This will make the review more complete.

Thank you for your constructive suggestion and we added a paragraph to narrate the role of technical advancement and mass spectrometer in identification PTMs from LH subtypes. I would like to mention that authors don’t have technical expertise in mass spec. and the mass spec. instrumentation.

Reviewer 2 Report

In this paper, authors summarize post-translation modifications and mutations in Linker histone, and their relationship with human diseases. Suggesting that targeting Linker histone may provide novel approach for diseases’ therapy. I think this paper could be accepted for publication after authors address question below.

1.     Authors should also summarize if there are small molecular inhibitors available for Linker histone modifications or Linker histone. As we know, there are many inhibitors for H3 methylation.

2.     So far, have ever researchers tried to target Linker histone modification or Linker histone for diseases’ therapy? No matter in vitro or in vivo. If yes, please also summarize them in this paper.

3.     In the section “Mutations in Linker histones and their implications”, authors mentioned that Linker histone mutation is associated with cancer. So whether Linker histone mutation affect survival of cancer patient? And what kinds of mutation primarily promote tumor growth.

4.     Except cancer. Diabetes, neurodegenerative disease and Cardiovascular disease are also major diseases threatening human heath. So whether Linker histone modifications and mutations are associated with these diseases as well. As I see authors just focus on cancer in this paper, which is inconsistent with the paper title.

Author Response

We thanks the reviewer for their constructive inputs. The typo error are corrected and highlighted with red font in the revised manuscript. In addition the edited sections are in blue color.

Reviewer#2

In this paper, authors summarize post-translation modifications and mutations in Linker histone, and their relationship with human diseases. Suggesting that targeting Linker histone may provide novel approach for diseases’ therapy. I think this paper could be accepted for publication after authors address question below.

We are highly thankful to the reviewer for their time and curiosity in multiple sections of the manuscript and providing their valuable suggestions. We tried to incorporate the suggestions as per best of the availability of information and as per best our understanding.

  1. Authors should also summarize if there are small molecular inhibitors available for Linker histone modifications or Linker histone. As we know, there are many inhibitors for H3 methylation.

A section entitled “Modulators of linker histone post-translational modifications” is added to address the reviewer suggestion.

  1. So far, have ever researchers tried to target Linker histone modification or Linker histone for diseases’ therapy? No matter in vitro or in vivo. If yes, please also summarize them in this paper.

The section entitled “Linker histone PTMs in disease” is especially summarizing the role of LH PTMs in disease (cancer, AD, PD and cardiovascular). Although, there are more studies in the context of cancer than other diseases. In addition, in the section entitled “Mutations in Linker histones and their implications” we already summarized the mutations which have a role in a rare congenital disease (Rahman syndrome) beside the mutations reported in cancer patient samples. There are few review articles which already mentioned the role of LH as an antimicrobial agent, in organ injury and disease (please refer to the recent review by Xia Li; https://pubmed.ncbi.nlm.nih.gov/36479112/; review by Eleanor Silk, 2017; https://pubmed.ncbi.nlm.nih.gov/28542146/; review by Paula S Espino , 2005, https://pubmed.ncbi.nlm.nih.gov/15723344/). We think mentioning about the linker histone in disease as such will distract the readers and will dilute the primary aim of this manuscript. Our objective in this manuscript is to highlight the importance of LH PTMs or LH mutations in disease.

  1. In the section “Mutations in Linker histones and their implications”, authors mentioned that Linker histone mutation is associated with cancer. So whether Linker histone mutation affect survival of cancer patient? And what kinds of mutation primarily promote tumor growth.

We thank reviewer for their curiosity on this aspect and we would like to mention that missesne substitutions in linker histone (LH) are found to be the most prevalent mutations from cancerous patient samples as mentioned in the COSMIC database and shown by pie chart in Figure 2. In best of our understanding and available information mutations in linker histone have a role in cancer as the identified mutations are obtained after whole genome sequencing from the multiple patients with cancers (COSMIC database) which suggests these mutation in LH might be driving the cancer possibly because of the improper function of LH (Mutated LH may not compact the chromatin and might be the oncogenes gets actively transcribed because of euchromatin state). The mechanistic role of these mutations or high frequency mutations (i.e. the amino acid residues which undergoes multiple mutations) in linker histones need to be explored and can be of great interest in the same way like scientific community is exploring the role of mutations in chromatin remodeling complexes (https://pubmed.ncbi.nlm.nih.gov/31759698/). We did not extrapolated the high frequency mutations from the database in this manuscript because the possibility of change in position of amino residues for high frequency mutation with the update in the database cannot be ruled out.

  1. Except cancer. Diabetes, neurodegenerative disease and Cardiovascular disease are also major diseases threatening human heath. So whether Linker histone modifications and mutations are associated with these diseases as well. As I see authors just focus on cancer in this paper, which is inconsistent with the paper title.

We agree with the reviewer that besides cancer there are multiple life threatening diseases but unfortunately there are very limited or no studies in the context of linker histone PTMs in these diseases. Although after going through extensive literature search we added a section of suggested role of linker histone PTMs in neurodegenerative diseases (Alzheimer and Parkinson disease). The extracellular histone has been explored in platelets activation to understand the blood coagulation and also in neutrophil extracellular traps (NETs), and sepsis but again these studies are not in context of human LH PTMs and our aim to keep this manuscript to address the human LH PTMs, mutations and their role in disease. To the best of our understanding we mentioned the title very carefully by using the word disease which stands generally for disease state instead of diseases which meant for multiple diseases. In any case if the reviewer is not satisfied with this we are flexible in this perspective (title).
